# OpenReview forum: "Improving Diversity of Demographic Representation in Large Language Models via Collective-Critiques and Self-Voting"
_EMNLP/2023/Conference — EMNLP 2023 Main_

### Official Review · Reviewer_3GqZ · 2023-07-30

**Soundness:** 3

**Excitement:**

3: Ambivalent: It has merits (e.g., it reports state-of-the-art results, the idea is nice), but there are key weaknesses (e.g., it describes incremental work), and it can significantly benefit from another round of revision. However, I won't object to accepting it if my co-reviewers champion it.

**Paper Topic And Main Contributions:**

This paper proposes collective critiques and self-voting to improve the diversity of generations in LLMs.
To evaluate this method, the authors propose a metric on people diversity and a dataset. Experiment results show that the proposed method can improve the diversity of generated text covering different genders and ethnicities.

**Reasons To Accept:**

1. The studied problem in this paper is important to the LLM generation. Diversity is a less well-defined problem in natural language generation. Maybe this paper can provide helpful settings for diversity evaluation, especially for LLMs.

2. The performance compared to other baselines is promising. The proposed method can effectively improve the diversity of the dataset.

3. This paper finds some interesting empirical results on LLM prompts for this problem, e.g., zero-shot prompting achieves similar or higher performance compared to few-shot prompting but it's more robust. These findings can be helpful for the following researches on LLMs.

**Reasons To Reject:**

1. The writing quality can be improved. E.g., Figure 2 is blurry.
2. The paper just studies the diversity of people and cities. I think it's a very narrow scope for natural language generation. Diversity can be measured in different aspects e.g., sentence patterns. But this paper focuses on entities in a generation (and only people and cities). Hence, I think the authors overclaim that this approach can improve the "diversity" of generation in LLMs.
3. Flan-PaLM 540B is a very specific model (and closed) for this problem and it's very huge. I think authors should include at least one open-source model for future duplication and at least one small-size model (e.g., alpaca or vicuna) to validate the model can be generalized to different models.

Overall, I think this is a good research problem for LLMs and the method is promising based on the experiments. Diversity is hard to measure and authors give a feasible metric for this problem. However, the experiments in this paper cannot be replicated (because of the closed model) and the proposed dataset is too narrow to validate the proposed method can improve "diversity".

**Reproducibility:**

2: Would be hard pressed to reproduce the results. The contribution depends on data that are simply not available outside the author's institution or consortium; not enough details are provided.

**Reviewer Confidence:**

3: Pretty sure, but there's a chance I missed something. Although I have a good feel for this area in general, I did not carefully check the paper's details, e.g., the math, experimental design, or novelty.

---

> ### Author Rebuttal · Authors · 2023-08-28
>
> We thank Reviewer 3GqZ for their thoughtful comments. We are encouraged that they found the studied problem important, and that they hope this paper can provide a helpful setting for diversity evaluation for LLMs. We are glad that they found our results interesting, comparison with baselines promising and diversity improvements effective, and our empirical findings helpful for following research on LLMs (Taken from Reviewer 3GqZ Reasons to Accept).
>
> We address reviewer comments below and will incorporate all feedback.
>
> > The writing quality can be improved. E.g., Figure 2 is blurry.
>
> Thank you for pointing this out, we will regenerate and update Figure 2 in the revision. Please feel free to make any other writing quality suggestions, we will do our best to address all of them in the revision, as well as take another pass on the text ourselves.
>
> > The paper just studies the diversity of people and culture. I think it's a very narrow scope for natural language generation. Diversity can be measured in different aspects e.g. sentence patterns …
>
> We are sorry the title wasn’t clear on “demographic diversity” vs “general diversity” (e.g., sentence patterns); we will update the paper to make this clearer. Our focus in this work is on “fairness and inclusion concerns in LLMs caused due to lack of diversity of representation of various demographic groups” (lines 43-45). Improving and evaluating “general diversity” is beyond the scope of this paper. We apologize for not stating the scope sufficiently clearly. We will revise parts of the abstract, introduction and related work to clarify this further. We will also update the title of the paper to “Improving Diversity of Demographic Representation ...”.
>
> While we understand that this does not cover all forms of diversity (e.g., sentence patterns), we believe that improving people and culture diversity in open-ended generation in large language models is an important and open research problem in its own right.  A breadth of past work has pointed out the practical importance of exposing diverse demographics and cultural contexts [1,2], and a large amount of research has focused on how to solve this problem in other fields like search and recommender systems [3,4,5,6].  However, this has remained largely an open problem in text generation. As a result, we believe studying this problem is important and developing fundamental solutions is valuable.
>
> Furthermore, prior work on diversity in other domains such as ranking and recommendation has shown that effective technical solutions for improving general diversity often fail to improve people and culture diversity [3,4,5]. Therefore alongside approaches to improve general diversity in LLMs, it is crucial to investigate approaches specifically for improving people and culture diversity, if we want to make progress on this important problem.
>
> > Flan-PaLM 540B is a very specific model (and closed) for this problem and it's very huge. I think authors should include at least one open-source model and at least one small-size model …
>
> We will extend our experiments analysis to include results on the follow up PaLM2 model which is an open model accessible via API. We will also try to include results on a smaller model.
>
> References:
> * [1] Unequal Representation and Gender Stereotypes in Image Search Results for Occupations. (CHI ’15).
> * [2] Media and the representation of Others. (International Social Science Journal 2010).
> * [3] Implicit diversity in image summarization. (HCI’ 2020)
> * [4] Representation online matters: Practical end-to-end diversification in search and recommender systems (FAccT’23)
> * [5] Fairness of Exposure in Rankings (KDD’18)
> * [6] Generating and controlling diversity in Image search (WACV’22)

---

### Official Review · Reviewer_wBnm · 2023-08-04

**Typos Grammar Style And Presentation Improvements:** 1. Authors should cite self-consisten…
**Soundness:** 4

**Excitement:**

4: Strong: This paper deepens the understanding of some phenomenon or lowers the barriers to an existing research direction.

**Missing References:**

1. Not a missing reference per se, but authors might want to look into "Let's Sample Step by Step: Adaptive-Consistency for Efficient Reasoning with LLMs" (Aggarwal et al., 2023) as future work to minimize the cost of CCSV.

**Paper Topic And Main Contributions:**

This paper focuses on the problem of diversity representation in open-ended LLM-generated text. It extends the idea of self-consistency (Wang et al., 2022) to propose a novel 0-shot method called Collective-Critiques and Self-Voting (CCSV). Authors evaluate both 0- and 5-shot versions of this method on a new dataset centered on gender and ethnic diversity across 105 professions, using automated measures --- "entropy" and "max-gap" for distribution- and extrema-wise evaluation, respectively --- as well as a careful selection of human judges. Authors include relevant baselines and an independent domain centered on cultural diversity. They also test annotator agreement, settings in which diversity requirements coexist with user-specified constraints, and ablated contributions within CCSV. Their findings favor CCSV both in the automated and human evaluations.

**Questions For The Authors:**

1. What do authors think about the cost structure of CCSV versus the baselines (see Reasons to Reject, #1)? This should be clarified in the limitations.
2. Do authors have early findings if CCSV works in other languages other than English? It can be mentioned as a limitation.
3. Is the temperature for the baselines (e.g., 0-shot IF) also 0.7?

**Reasons To Accept:**

1. This is a well-written paper about an important topic, especially with the rise of LLMs in open-ended generation tasks.
2. The method builds on state-of-the-art work and is thoroughly evaluated, i.e., evaluations include meaningful baselines and metrics, both automated and human-based --- with proper consideration of judges' diversity --- and over two independent domains.
3. Findings support authors' claims that CCSV appears to induce greater diversity, be more robust to different domains, and be more flexible to user-specified constraints, while requiring less task-specific engineering.

**Reasons To Reject:**

1. While I would not consider this a strong reason for rejecting the paper in its current state, it is still an important limitation that authors' analyses do not include the cost dimension. It would be important for the practical usefulness of CCSV to understand how many more tokens the sampled critiquing and voting steps cost compared with baseline methods --- especially standard 5-shot. Authors mention that CCSV reduces the number of iterations, but this is not a good proxy for cost in this case given all the additional decodings. There are other dimensions favoring CCSV --- most importantly, greater independence to task-specific engineering and greater robustness --- but it would useful and fair to understand the potential downside in cost.
2. According to line 469, the temperature for the baselines (e.g., 0-shot IF) is also 0.7. However, baselines based on a single decoding should use zero temperature. Using non-zero temperature, it would be fair to include CoT with self-consistency as a baseline.

**Reproducibility:**

3: Could reproduce the results with some difficulty. The settings of parameters are underspecified or subjectively determined; the training/evaluation data are not widely available.

**Reviewer Confidence:**

4: Quite sure. I tried to check the important points carefully. It's unlikely, though conceivable, that I missed something that should affect my ratings.

---

> ### Author Rebuttal · Authors · 2023-08-28
>
> We thank Reviewer wBnm for their insightful and positive feedback! We are encouraged that they found the problem of diversity of representation in open-ended LLM generations an important research problem, our paper well-written, our method thoroughly evaluated with meaningful baselines and metrics our human-based evaluation carefully designed, and that our empirical findings on diversity and robustness experiments supported the claims (Taken from Reviewer wBnm Reason to Accept).
>
> Thanks also for the thoughtful comments and suggestions for future work. We ran additional experiments to answer the specific questions below, we will incorporate all feedback in the revision.
>
>
> > What do authors think about the cost structure of CCSV versus the baselines? This should be clarified in the limitations.
>
> Thank you for highlighting this. We will update the discussion section of the paper with a cost analysis, and expand limitations section to include a discussion of inference cost. Summarizing briefly the cost structure, there are two pieces that contribute to the cost of CCSV: (i) Number of Iterations and (ii) Number of additional output tokens generated for critique and revision step.
>
> On (i) number of iterations, it is worth highlighting that, while CCSV is an iterative method, in practice we observed substantial gains already after 1 round of interaction. In fact, all the results in the experiment section are reported after only 1 iteration. Further, when compared to vanilla greedy-critiquing (used in state-of-the-art baseline CAI), our proposed collective-critiquing step achieves similar gains in fewer iterations, thus improving cost-diversity trade-off (see Fig. 4).
>
> We also thank Reviewer wBnm for pointing us to relevant papers for future work for minimizing the cost of CCSV; we will add these citations. Beyond their suggested “efficient reasoning” methods, another future work for minimizing the cost would be to use the CCSV method in an offline fashion to generate better synthetic supervised data to fine-tune the model, such that the improved model can give more diverse predictions in a single inference run after fine-tuning.
>
> We will discuss (ii) cost of generating additional output tokens in the next point below.
>
> > How many more tokens the sampled critiquing and voting steps cost compared with baseline methods --- especially standard 5-shot.
>
> In contrast to our proposed CCSV approach, which is zero-shot, the few-shot baselines need substantially more input tokens. For simplicity of analysis, let us treat each word as a token. The few-shot exemplars for standard 5-shot prompting baseline take ~350 additional tokens (See A.4 Tbl 6 in supplementary). On the other hand, CCSV needs more output tokens for decoding critique and revision. On average, the critique step needed additional ~25 tokens, and the revision step needed ~30 tokens. (See A.1 Fig. 5 in supplementary for an example of critiques and revisions decoded). Thus, while we agree that CCSV needs additional output tokens, we believe the additional cost in decoding is not substantial, especially as we see substantial diversity improvements already after 1 round of critique and revision.
>
> > Do authors have early findings if CCSV works in other languages other than English? It can be mentioned as a limitation.
>
> Thank you for the suggestion. Indeed CCSV was only tested in English. We will expand our limitations section to discuss this and highlight the need for future work to tackle the problem of diversity and inclusion in a multilingual setup. We ran a quick experiment by manually translating a few of the queries from our eval dataset into Chinese, and running the baseline vs CCSV method as-is, i.e., without making any changes to the prompting technique or the prompt text for critique and revision (which remained in English). Anecdotally, we observed that in most cases the CCSV approach was able to critique the base response (which was in Chinese) correctly, and revise the draft to improve diversity. However, as expected due to the English instructions, both the generated critique and revision were in English. We leave the investigation of multilingual CCSV for future work, and will mention this in the paper.
>
> > Is the temperature for the baselines (e.g., 0-shot IF) also 0.7?
>
> Yes, for fairness of comparison we report all baselines and methods at the same temperature. We use temp=0.7 for all methods as it is typical for production models (e.g., ChatGPT, PaLM2) to run at non-zero temperature. That said, we performed additional experiments at temperature 0.0 for the Baseline, 0-shot CoT, and 0-shot IF methods. We see a slight improvement in the diversity scores at temperature 0.0 – e.g., gender gap for Baseline reduces from 0.99 to 0.98, for 0-shot IF from 0.99 to 0.97, 0-shot CoT from 0.99 to 0.97 – but the overall trends remain the same.
>
> > Using non-zero temperature, it would be fair to include CoT with self-consistency as a baseline.
>
> The self-consistency method can only be applied to problems where the final answer is from a fixed answer set, and does not extend easily to open-ended text generation. Thus we only compare with the baseline CoT method. In fact, the authors of the self-consistency paper [1] highlight this as a known limitation and call out an extension to the open-text generation problem as a direction for future work (Page 4, last paragraph of their related work.). Our proposed CCSV approach can in some ways be seen as extending the self-consistency method to open-ended generations.  We’ll update the related work and experiments section of the paper with a brief note on this.
>
> > Misc, Typo and Presentation Improvements
>
> Thank you for the suggested citation, typo fixes, and suggested presentation improvements. We will address all the suggestions in the revision.
>
> References:
>
> * [1] Self-Consistency Improves Chain Of Thought Reasoning In Language Models [Wang et al. 2023]

---

### Official Review · Reviewer_18RZ · 2023-08-05

**Soundness:** 3

**Excitement:**

4: Strong: This paper deepens the understanding of some phenomenon or lowers the barriers to an existing research direction.

**Paper Topic And Main Contributions:**

This paper tackles the challenge of limited diversity in LLM responses along the cultural and ethnic dimensions, especially wrt. people questions - e.g., name x people who achieved a task, or occupied a position such as CEO of a major corporation. Models typically exhibit bias towards the majority populations in the training datasets and thus need specialized techniques to mitigate the bias. This paper proposes a self-critiquing/prompting technique to address this challenge. Specifically, the model is prompted to identify its biases and subsequently asked to correct its own biases.

I thank the authors for their detailed rebuttal responses to my concerns. However, I still have some concerns wrt. reproducibility, and strongly advice the authors to open source their code in future submissions during the review stage. On the whole, I retain my scores - Strong excitement, good soundness, some concerns wrt. reproducibility. I hope the authors find my suggestions helpful to improve their work.

**Questions For The Authors:**

I request the authors to fully open source their code and experimental infrastructure as well as the datasets given the importance of this problem area.

**Reasons To Accept:**

1) Important problem area - With the increasing adoption of LLMs, a large proportion of the content available on the internet may be party or entirely generated via LLMs. In such a scenario, it is important to ensure diversity in the content that internet users consume and provide exposure to diverse viewpoints and cultural contexts.

2) The proposed technique is simple and elegant - In theory, any model exposed to a large volume of data should be able to self critique. Thus the proposed should work well for large LLMs trained on trillions of tokens.

3) Does not make any architectural assumptions and hence applicable to any LLM with any architecture and inference setup.

**Reasons To Reject:**

1) Limited reproducibility - I request the authors to both, open source their code, and release their datasets, including the modified versions. Typically, an anonymized code/data repository helps reviewers verify the claims of the paper.

2) Assumes the availability of diverse knowledge in the LLM - Typically an LLM can only self-critique if it is exposed to alternative forms of responses in the training dataset. For smaller LLMs trained with smaller volumes of data, they may have limited exposure to diverse content and hence, incapable of self-critique.

3) This method is somewhat computationally expensive - For a typical response, the authors engage in iterative improvement via multiple rounds of self critiquing. This may prove expensive for large LLM models, computationally and in terms of user response time in real time scenarios such as chat bots.

**Reproducibility:**

3: Could reproduce the results with some difficulty. The settings of parameters are underspecified or subjectively determined; the training/evaluation data are not widely available.

**Reviewer Confidence:**

3: Pretty sure, but there's a chance I missed something. Although I have a good feel for this area in general, I did not carefully check the paper's details, e.g., the math, experimental design, or novelty.

---

> ### Author Rebuttal · Authors · 2023-08-28
>
> We thank Reviewer 18RZ for their thoughtful feedback and suggestions. We are encouraged that they find the problem area important, our proposed technique simple and elegant. We are pleased that they appreciate our design  which does not make any architectural assumptions and hence can be applicable to any LLMs (From Reviewer 18RZ’s Reasons to Accept).
>
>
> We address reviewer comments below and will incorporate all the feedback in the revision.
>
>
> >  I request the authors to release their datasets, including modified versions …
>
> The supplementary sections sec. A.2.1 and A.2.2 on “Implementation of Evaluation Datasets” already include all the hand-crafted templates, as well as the terms used to populate the templates in order to construct the datasets. In addition, we will be releasing all the datasets (including the modified datasets used in ablation studies) as csv files on Github. Indeed, our hope is that the released datasets will be used in future work to benchmark models for diversity of demographic representation.
>
>
> >  I request the authors to open source their code…
>
> The supplementary section A.3 on “Implementation of Baseline and Methods” reports the exact in-context prompt text used for each of the baseline methods, including the few-shot exemplars used (sec A.4). Figure 2 visualizes the pseudo-code for CCSV, with the exact prompts used for each step. In addition, we are working on releasing the model-abstracted code for our CCSV method, as well as the implementation of diversity metrics.
>
>
> > smaller LLMs trained with smaller volumes of data, they may have limited exposure to diverse content and hence, incapable of self-critique.
>
> Indeed extending such capabilities to smaller models is an important and open research problem. However, we believe that even if it turns out that only large models are inherently able to understand diversity and generate diverse responses, this would still be a generally useful technique that can benefit a wide variety of models.  For example, one direction for future work would be to leverage CCSV in an offline setup to generate better (more diverse) synthetic supervised data using larger LLMs, and use this data to “teach” small language models via fine-tuning the smaller “student” models. Similar approaches have been applied in the past to “teach small language models to reason” via knowledge-distillation [1].
>
>
> > This method is somewhat computationally expensive …
>
> We agree one limitation of CCSV is that it incurs more computation cost (much like any other iterative reasoning method, including Constitutional AI). However, it is worth highlighting that, while CCSV is an iterative method, in practice we observed substantial gains already after 1 round of interaction. In fact, all the results in the experiment section are reported after only 1 iteration (see line 255). Further, when compared to vanilla greedy-critiquing (used in state-of-the-art baseline CAI), our proposed collective-critiquing step achieves similar gains in fewer iterations, thus improving cost-diversity trade-off (see Fig. 4). As part of future work, one could use the CCSV method in an offline fashion to generate better synthetic supervised data to fine-tune the model, such that the improved model can give more diverse predictions in a single inference run after fine-tuning.
>
> References:
> * [1] “Teaching Small Language Models to Reason” [Magister et al. 2023]

---

### Meta-Review · Area_Chair_imm8 · 2023-09-16

**Recommendation:** 5

**Metareview:**

This paper looks at the diversity of outputs generated by LLMs. They note that generation tends to lack diversity when it comes to text generated from underrepresented groups. They develop a new metric to measure diversity and then present a prompting technique to improve on this metric. Overall, the reviewers found this work to be sound and exciting. This is a very important application area. Two of the reviewers mentioned a cost analysis that would be helpful to be included. They reviewers also mentioned the need for some open source code/models to be able to reproduce the study. They also mention important limitations to discuss in the paper including the type of diversity, cost, and access to diverse knowledge. Even with these limitations, this paper makes a solid contribution with thorough automatic and human evaluation.

---

### Decision · Program_Chairs · 2023-10-07

**Decision:**

Accept-Main

**Comment:**

This paper looks at the diversity of outputs generated by LLMs. They note that generation tends to lack diversity when it comes to text generated from underrepresented groups. They develop a new metric to measure diversity and then present a prompting technique to improve on this metric. Overall, the reviewers found this work to be sound and exciting. This is a very important application area. Two of the reviewers mentioned a cost analysis that would be helpful to be included. They reviewers also mentioned the need for some open source code/models to be able to reproduce the study. They also mention important limitations to discuss in the paper including the type of diversity, cost, and access to diverse knowledge. Even with these limitations, this paper makes a solid contribution with thorough automatic and human evaluation.